# Atrial Fibrillation after Lung Cancer Surgery: Prediction, Prevention and Anticoagulation Management

**DOI:** 10.3390/cancers13164012

**Published:** 2021-08-09

**Authors:** Gennaro Carmine Semeraro, Carlo Ambrogio Meroni, Carlo Maria Cipolla, Daniela Maria Cardinale

**Affiliations:** 1Cardioncology Unit, European Institute of Oncology, IRCCS, 20145 Milan, MI, Italy; daniela.cardinale@ieo.it; 2Cardiology Department, European Institute of Oncology, IRCCS, 20145 Milan, MI, Italy; CarloAmbrogio.Meroni@ieo.it (C.A.M.); carlo.cipolla@ieo.it (C.M.C.)

**Keywords:** anticoagulation, amiodarone, beta-blockers, lung cancer, NT-proBNP, postoperative atrial fibrillation, thoracic surgery, surgery complications

## Abstract

**Simple Summary:**

Atrial fibrillation that occurs after surgery raises further questions with respect to spontaneous atrial fibrillation, being an event unquestionably related to the surgical act itself and always quite self-limiting. The purpose of this review is to present the knowledge gained so far, including the most recent findings, regarding this peculiar form of arrhythmia. Its prognostic impact and the possibility of predicting and preventing it were the subject of our analysis, as well as the similarities and differences with spontaneous atrial fibrillation in relation to anticoagulation. Where possible, the search for evidence has focused on studies involving lung cancer patients undergoing thoracic surgery, highlighting any differences with cardiac surgery.

**Abstract:**

Atrial fibrillation (AF) is a common complication of the early postoperative period of various types of surgery, including that for lung cancer. Although induced by the homeostatic alterations related to surgery, there is evidence that it is not a mere stand-alone transitory event, but it represents a relevant complication of surgery, bearing considerable prognostic consequences. Different methods have therefore been explored to predict the occurrence of postoperative atrial fibrillation (POAF) and prevent it. In particular, the age among clinical factors, and N-terminal prohormone of brain natriuretic peptide (NT-proBNP), as a marker, have proven to be good predictors, and the use of beta-blockers or amiodarone in primary prevention seems to reduce its incidence significantly. There is growing evidence that POAF significantly increases the risk of stroke and global mortality in the long term; therefore, it should be managed in the same way as spontaneous atrial fibrillation. In this review, we will present the strongest evidence found so far and the most recent findings regarding the management of POAF, with a special focus on patients undergoing thoracic surgery for lung cancer.

## 1. Introduction

Atrial fibrillation (AF) has a particularly high incidence in cancer patients. This is due to various reasons: the dysregulation of the immune system homeostasis in a pro-inflammatory sense, the potential arrhythmogenicity of some chemotherapeutic agents, the impact of surgery when performed, the possible cardiac infiltration of cancer cells [1,2].

In the oncologic setting, in addition to the standard clinical dilemmas related to the management of this arrythmia, further questions may arise, and this is even truer when approaching postoperative atrial fibrillation (POAF), which often complicates thoracic surgery.

POAF, in fact, due to its close temporal correlation with the surgical intervention, is not generally thought of as a stand-alone pathology but rather as an event triggered by surgery-induced stress, usually self-limiting and transitory. Accordingly, it is not so easy to establish whether it should be approached in the same way as spontaneous AF or whether it should be managed in a different way. In particular, the need has arisen to understand its prognostic impact, to investigate the possibility of preventing it with appropriate strategies and find a way to predict its occurrence in order to reserve the appropriate preventive interventions only for patients at high risk.

In fact, it is precisely due to its strict correlation with surgery and thus being treated simply as a direct consequence of it that in the past, POAF was mistakenly considered a quite harmless complication. However, over time, it has become clearer how it exerts a negative impact on patient prognosis, extending hospital stay, and increasing the risk of stroke and overall mortality [3].

The purpose of this review is to present the strongest evidence found so far and the most recent findings regarding the management of POAF, with a special focus on patients undergoing thoracic surgery for lung cancer.

## 2. Incidence

Depending on the type of surgery, POAF complicates between 20 and 40% of cardiac operations and between 10 and 20% of non-cardiac operations [4]. A study by Roselli et al. showed that in patients undergoing thoracic surgery for lung cancer, in particular, the incidence was 19% [5]. Moreover, the incidence varies according to the extent of lung resection, being lower for wedge resections (2–4%), intermediate for lobectomies (10–15%), high for pneumonectomies (>20%). Most cases occur between the 2nd and 4th postoperative (PO) day [6] (Figure 1).

Ivanovic et al. showed that 23.3% of patients undergoing non-cardiac thoracic surgery develop POAF within the first 24 h. The peak of incidence occurs approximately at PO day 2.5, while by the 3rd PO day, 60% of the episodes of the total recorded had already occurred. Only 11% were found to experience POAF beyond the 5th PO day [7].

In a study by Higuchi et al., about 80% of the patients experiencing POAF had no symptoms [8]. This would imply that a significant percentage of POAF episodes could remain undetected if patients are not undergoing continuous postoperative electrocardiographic recording.

## 3. Phatophysiology

It is widely accepted that POAF is triggered by multiple factors. Some of these are pre-existing, while others can be induced by surgery-related stress and are usually transitory in their nature. Leaving aside the underlying predisposing pathological substrate (i.e., atrial wall fibrosis, some valvulopathies, and other forms of heart disease of which the patient may already be a carrier), the main homeostatic alterations triggered by surgery and possibly provoking AF are adrenergic activation and systemic inflammation. In fact, it has been shown that high postoperative norepinephrine values, as well as the use of vasopressors or inotropes, are related to a greater risk of developing POAF, while the use of beta-blockers for prophylactic purposes reduces its incidence [4].

Direct damage to the nerve structures of the heart during thoracic surgery has also been posited as a potential explanation for the autonomic changes following surgery [9].

In addition, hypo/hypervolemia, anemia, hypoglycemia, and plasma electrolyte changes can alter the sympathetic tone and can trigger POAF. In the case of lung surgery, the vasoconstriction of the pulmonary veins can produce a pressure increase in the right heart chambers with consequent stretching of the atrium [10].

Finally, a further contribution can be provided by the activation of inflammation. In addition to systemic inflammation, local inflammation of the structures adjacent to the resection site, or even inflammation of the resected pulmonary veins, can trigger POAF [11].

## 4. Prognostic Implications

Being strictly related to surgery, POAF has been considered in the past as a low-risk entity, limited to the postoperative phase. [12] Over time, however, new evidence has emerged showing that it has a high incidence of relapses and related complications and that it actually represents an unfavorable prognostic indicator. Several studies, in fact, have concluded that POAF is associated with an increased risk of stroke and mortality. In particular, recent meta-analyses have investigated precisely how much this risk increases:Lin et al. showed how POAF increases the risk of stroke and mortality by about one and a half times, both in the short and long term. In patients undergoing non-cardiac surgery, this risk was approximately doubled [13];Koshy et al. showed that the risk of stroke related to POAF was 2.5 times higher than in patients who did not experience this arrhythmia during a follow-up of an average duration of 1.4 years; this risk was three times higher in patients undergoing non-thoracic surgery [14];AlTurki et al. showed a threefold increase in short-term stroke risk and approximately fourfold mortality risk in the long-term [15];Albini et al. showed a fourfold increase in long-term stroke risk and about 3.6 times in long-term mortality risk [16] (Table 1).

Regardless of stroke incidence, Lowres et al. conducted a meta-analysis to investigate the incidence of AF relapse after the first episode of POAF. Since arrhythmic episodes in these patients are often asymptomatic, only studies that adopted non-invasive continuous electrocardiographic monitoring were included. The investigators found that the incidence of AF relapse was 28.3%. When continuous monitoring through implantable devices was employed, relapse was demonstrated to occur in 61–100% of the patients within two years [17].

## 5. Risk Factors and Predictors

### 5.1. Clinical

#### 5.1.1. Patient-Related

A meta-analysis by Yamashita et al. in 2019 on studies involving a total of 36,834 subjects found that in patients who developed POAF, compared to those unaffected, there was a significant difference in terms of mean age, mean atrial diameter, and mean left ventricular ejection fraction. The first two parameters were higher, and the last one was lower in patients suffering POAF. Heart failure (HF), arterial hypertension, chronic obstructive pulmonary disease (COPD), and a history of myocardial infarction significantly increased the risk [18]. In addition, male sex, obesity, previous episodes of AF, and poor performance status have been shown to be associated with an increased risk of POAF [4,6,19].

#### 5.1.2. Surgery-Related

The extent and type of thoracic surgery seem to affect the likelihood of developing POAF. Pneumonectomies and lobectomies generally carry a greater risk than wedge resections. In general, the overall volume of resected lung appears to be an important risk factor [6,20,21] (Table 2).

Regarding pneumonectomy, some investigators have reported left laterality as a potential risk factor, but not all studies are in agreement [21]. In addition, lymph node dissection in the context of lung surgery was found to be associated with a higher incidence of POAF in a prospective study carried out by Cardinale et al. in 1999 [3].

On the basis of the progressive greater use and refinement of minimally invasive surgery with video-assisted thoracoscopy, Seitlinger et al. recently investigated the differences in terms of outcomes and complications between patients treated with the aforementioned technique and patients scheduled to be treated with the same, but whose intervention needed conversion to traditional surgery for various reasons. In general, the latter had a significantly higher complication rate and, regarding the incidence of POAF, it was much higher in the group that required conversion. In accordance with literature data, the authors concluded that conversion may result in an even higher POAF occurrence than planned thoracotomy. Given the low conversion rate observed in centers with high expertise, they stressed the importance of surgeon training for reducing the incidence of such complications [22].

### 5.2. Biomarkers

#### 5.2.1. Natriuretic Peptides

One of the first studies to prospectively analyze the ability of natriuretic peptides to predict POAF was carried out in 2007 by Cardinale et al., who evaluated NT-proBNP 24 h before and 1 h after thoracic surgery for lung cancer in 400 patients: In patients with an increase in this marker, the incidence of POAF was much higher than in patients with normal values. Both the preoperative and postoperative values were able to predict the occurrence of POAF with a relative risk of 27.9 and 20.1, respectively [23] (Figure 2). This result was also confirmed by a subsequent study by the same group carried out on a larger sample of patients [24].

Several other studies have confirmed this finding. Nojiri et al., in 2009, analyzed the preoperative and postoperative brain natriuretic peptide (BNP) and atrial natriuretic peptide (ANP) values in patients undergoing pulmonary resection for lung cancer. The preoperative BNP value was found to be the most effective predictor of POAF (area under the ROC curve 0.90). A cut-off of 30 mg/mL was found to have a sensitivity of 77%, a specificity of 93%, a positive predictive value of 81%, and a negative predictive value of 92%. [25] Ata et al., in 2009, carried out a study on patients undergoing coronary artery bypass grafting (CABG), which showed that the preoperative BNP value was a predictor of POAF, with an area under the ROC curve of 0.75. A cut-off of 135 pg/mL showed 72.2% sensitivity, 71.2% specificity, 45.6% positive predictive value, and 88.5% negative predictive value for POAF [26].

Unlike other authors, Masson et al., in 2015, found that the pre-surgical NT-proBNP concentration or its variations pre- and post-surgery were not significantly associated with POAF. [27] in addition, Hernández-Romero et al. also found that NT-proBNP was not a good predictor [28].

From a meta-analysis by Cai et al., it emerged that the sensitivity and specificity of natriuretic peptides for POAF are 75% and 80%, respectively, with an odds ratio of 3.28. Moreover, NT-proBNP was shown to have a higher predictive capacity than BNP, as did the postoperative measurement compared to the preoperative one; BNP showed a better correlation with POAF in patients undergoing thoracic surgery than in those undergoing cardiac surgery. [29] A further meta-analysis by Simmers et al. from 2014 confirmed these findings [30].

Despite some contrast in evidence, in general, data gathered so far seem to indicate a good predictive capacity of natriuretic peptides, in particular NT-proBNP, for POAF.

#### 5.2.2. Troponin

Studies on troponin as a predictor of POAF have all been carried out in patients undergoing cardiac surgery. Knayzer et al., in 2006, found no correlation between the postoperative values of Troponin I (TnI) and AF occurrence [31]. Leal et al., in 2011, found that TnI dosed early after CABG was significantly higher in patients who had POAF than in those who remained in sinus rhythm. A cut-off value of 0.901 ng/mL showed a sensitivity of 60% and a specificity of 87% for POAF prediction [32]. Hernández-Romero et al., in 2013, demonstrated that high pre-surgical high-sensitive troponin T (HS-TnT) values were independent predictors of POAF, while post-surgical values were not good predictors, suggesting that intra- and postoperative myocardial injury is not associated with this arrhythmia [28]. Lahoz-Tornos, in 2014, found that low HDL cholesterol and high troponin T (US-TnT) values detected preoperatively could predict POAF, and that a US-TnT cut-off of 11.86 ng/L had a sensitivity of 76% and a specificity of 54% [33]. Finally, Masson et al. in the OPERA trial found no predictive capacity of hs-cTnT for POAF [27].

To our knowledge, no meta-analyses have been carried out to date to assess the real utility of troponin in predicting POAF.

#### 5.2.3. C Reactive Protein and Inflammation Markers

Given the relationship between increased systemic pro-inflammatory status and higher AF incidence, markers of inflammation were analyzed as possible predictors of POAF.

A meta-analysis by Li et al. in 2015 showed that in patients undergoing CABG for coronary artery disease, postoperative C-reactive protein (CRP) values were higher in those who had POAF in comparison to non-POAF patients. After stratifying by ethnicity, this finding was true only for white patients but not for Asian patients [34]. In 2017, Weymann et al. found that preoperative levels of CRP and interleukin (IL) 6 and postoperative levels of CRP, IL-6, IL-8, and IL-10 were significantly higher in patients with POAF after cardiac surgery [35].

More recently, in a retrospective study published in 2020 by Olesen et al., a higher C-reactive protein value in the postoperative period was associated with a higher incidence of POAF [36]. A meta-analysis by Liu et al. carried out in 2020 showed that a perioperative elevated neutrophil/lymphocyte ratio was associated with an increased risk of POAF in patients undergoing cardiac surgery [37].

#### 5.2.4. Postoperative Noradrenalin

As previously mentioned, the activation of the sympathetic system in the perioperative period contributes to triggering POAF. In order to demonstrate the possible role of the catecholamines in the genesis of this arrhythmia, Kalman et al. evaluated the association between postoperative plasma norepinephrine values and the incidence of POAF in patients undergoing CABG. These were actually higher in patients who experienced POAF [38].

### 5.3. Echocardiograpic Parameters

Several echocardiographic parameters have also been studied to understand their predictive capacity towards POAF predominantly in cardiac surgery patients. Studies on diastolic function have reported conflicting results. One of them by Nojiri et al. showed that left ventricular early transmitral velocity/mitral annular early diastolic velocity (E/e’) ratio had a sensitivity of 90% and a specificity of 73%, but this result was not subsequently confirmed [21,39]. The size and function of the left atrium were, for obvious reasons, also the subject of research. The anteroposterior diameter of the left atrium was shown to be positively correlated with the incidence of POAF in a study by Lin et al. [40]. Hidayet et al. carried out a study to analyze the association between 3D parameters for the evaluation of left atrial (LA) mechanical functions and POAF in patients undergoing CABG. A positive correlation was found between POAF and LA maximal volume, LA atrial precontraction volume, LA active stroke volume, LA expansion index, and LA volume index. Moreover, there was a negative correlation between LA total emptying fraction and LA passive ejection fraction [41]. A study by Pernigo et al. in patients undergoing aortic valve replacement for severe stenosis showed that peak atrial longitudinal strain and peak atrial contraction strain values were inversely correlated with this arrhythmia [42]. Hu et al., analyzed changes in GLS with transesophageal echocardiography before and immediately after surgery (T2) in patients undergoing aortic valve replacement. GLS (T2) and ΔGLS% were independent predictors of POAF [43].

### 5.4. Risk Scores

Various risk scores commonly used in clinical practice were also analyzed in relation to their ability to predict POAF, mainly in the context of cardiac surgery. As expected, most of the studies focused on CHA2DS2-VASc. A meta-analysis by Chen et al. incorporated many of those studies, and it emerged that CHA2DS2-VASc is an independent predictor of POAF after cardiac surgery. The higher was the patient’s score, the higher was the incidence of the arrhythmia [44] (Figure 3).

Lee et al., in 2019, studied the same score but specifically in patients who underwent pulmonary lobectomy. In addition, in this setting, CHA_2_DS_2_-VASc was found to be a good predictor of POAF. [45] Other scores analyzed were Syntax which, in a study by Geçmen et al., showed a good correlation with the incidence of POAF in patients undergoing CABG, and Hatch, which, in a study by Erdolu et al., was found to differ significantly in patients who then developed POAF compared to those who remained in sinus rhythm after CABG [46,47]. Mariscalco et al. even validated a specific score in 2014, termed the POAF score, based on the major predictors identified in over 17,000 enrolled patients [48]. A comparison among the various scores was made by Burgos et al., who demonstrated that CHAD2DS2-VASc had a greater discriminative ability to predict the event than did POAF and HATCH. However, all three scores were found to be independent predictors of postoperative atrial fibrillation [49].

## 6. Prevention

Given the impact in terms of morbidity, mortality, and costs of POAF, a series of prophylactic measures aimed at reducing its incidence have been analyzed; in particular, the preventive capacity of various pharmacological therapies to be administered before, during, and/or immediately after surgery has been the subject of study. Studies have been conducted both in patients undergoing thoracic surgery, specifically for lung cancer, and in patients undergoing cardiac or other surgery. From various meta-analyses, it has emerged overall that the application of a preventive strategy seems to yield promising results, reducing the risk of developing POAF after surgery by about 50% on average. It must be said, however, that not all the prevention methods investigated have shown equal effectiveness.

### 6.1. Beta-Blockers

The first studies on the prevention of postoperative AF with beta-blockers date back to the late 1990s.

#### 6.1.1. Metoprolol

Metoprolol was one of the first and most studied among beta-blockers for POAF prevention. In 1997 Jakobsen et al. showed that patients treated with metoprolol both before surgery for lung resection and postoperatively had a significantly lower POAF incidence compared with the untreated patients [50]. Metoprolol produced satisfactory results, in order to prevent POAF in patients undergoing cardiac surgery and also in a study by Skiba et al., albeit with a high incidence of side effects, in particular, bradycardia [51]. Cardinale et al. investigated the preventive capacity of this drug, as well as that of losartan, in the Presage trial. Only patients with high NT-proBNP values detected 24 h pre-surgery or 1 h after it, were randomized to receive one or the other drug or no therapy: The incidence of POAF in pharmacologically-treated patients was significantly lower than the incidence in the control group. On the other hand, no difference in terms of preventive capacity was found between the two therapies [24]. A recent meta-analysis by Norhayati et al. on patients undergoing cardiac surgery confirmed the efficacy of metolprolol [52].

#### 6.1.2. Other Long-Acting Beta-Blockers

Given the proven efficacy of metoprolol, other beta-blockers with theoretically more favorable pharmacodynamic properties for POAF prevention were tested in order to verify their potential greater efficacy and/or their capacity to induce fewer side effects.

Carvedilol significantly reduced the incidence of POAF and was more effective than metoprolol in patients undergoing cardiac surgery, as emerged in a meta-analysis by Wang et al. [53].

Sotalol, also having class 3 antiarrhythmic properties, has been assumed to be superior to other beta-blockers. In a study by Parikka et al. in patients undergoing CABG, it was superior to metoprolol in reducing the incidence of POAF [54]. A meta-analysis by Kerin et al. investigated the efficacy of sotalol in comparison with placebo, other beta-blockers, and amiodarone in reducing POAF in patients undergoing cardiac surgery. Sotalol was more effective than placebo and other drugs of the same class but not superior to amiodarone. The timing of administration was also relevant, given that in patients to whom sotalol was administered after surgery, the incidence of adverse events was lower than in those in which it was administered before surgery [55].

#### 6.1.3. Short-Acting Beta-Blockers

One of the main issues that limits the use of beta-blockers for preventing POAF is the fear that they may cause bradycardia, hypotension and have a negative inotropic effect in the perioperative period; they also counteract the action of cardiotonic and vasoconstrictive amines that may need to be administered during surgery. Furthermore, especially in patients with pneumological diseases, and therefore also those with lung cancer, they might cause bronchospasm.

Landiolol administered intravenously has a very short half-life, similar to that of esmolol, but has greater β1 selectivity (cardioselectivity) compared to the latter; given its quick clearance, any side effects related to its use can promptly regress after interrupting the infusion. Studies aimed at determining the capacity of this drug to prevent POAF have involved both patients undergoing cardiac surgery and patients undergoing thoracic surgery; results confirmed landiolol’s effectiveness in reducing POAF in the heart surgery setting. However, one study failed to prove any benefit in patients undergoing thoracic surgery [56,57,58]. A recent meta-analysis by Walter et al. comparing landiolol to the standard of care (i.e., pooled control group treated with other beta-blockers or diltiazem or no prevention), showed the greater efficacy of the former, which also contributed to a significant reduction in hospital stay costs [59].

#### 6.1.4. Overall Beta-Blockers Effectiveness

A recent meta-analysis by Kim et al. on beta-blocker-based POAF prevention proved their effectiveness but did not show any overall improvement in the clinical outcomes in patients who underwent CABG [60]. Of notable relevance was the finding of increased mortality in patients pre-treated with beta-blockers undergoing non-cardiac surgery in a meta-analysis by Oesterle et al. [61].

### 6.2. Amiodarone

Amiodarone has been studied to the same extent as beta-blockers for its preventive POAF effectiveness. Riber et al. carried out two studies involving lung cancer patients undergoing thoracic surgery. Amiodarone was able to significantly reduce the incidence of POAF but not the hospital stay length and costs [62,63]. Giri et al. investigated the ability of amiodarone to reduce POAF in patients already on beta-blockers undergoing cardiac surgery. The drugs combination proved to be better than placebo plus beta-blockers [64]. A meta-analysis by Zhu et al. found, on the contrary, that amiodarone was not superior to beta-blockers in reducing POAF occurrence in patients undergoing cardiac surgery [65].

### 6.3. Calcium Channel Blockers

A study on verapamil by Van Mieghem et al. showed its superiority to placebo in reducing POAF after lung operations; however, the drug induced side effects in a sizeable number of patients [66]. Diltiazem is included among the preventive strategies listed in the AATS “*Guidelines for the prevention and management of peri-operative atrial fibrillation and flutter (POAF) for thoracic surgical procedures*” for patients at intermediate risk of developing POAF [20]. Despite the indications set out in the guidelines, the results of recent studies led to substantially reconsidering the usefulness of calcium channel blockers. Hochreiter et al., in 2020, found that diltiazem did not reduce the incidence of POAF following thoracoabdominal esophagectomy. At the same time, its use was associated with several side effects [67]. Lederer et al., in a very recent study, showed that diltiazem prophylaxis was not useful in reducing POAF incidence after thoracoscopic lobectomy [68].

### 6.4. Renin-Angiotensin-Aldosterone System (RAAS) Inhibitors

Regarding RAAS inhibitors, the results of several studies were discordant. In a randomized study by Ozaydin et al., angiotensin-converting enzyme inhibitors (ACEI) reduced POAF incidence compared to placebo in patients undergoing cardiac surgery [69]. In the Presage trial by Cardinale et al., losartan showed efficacy in preventing POAF equivalent to that of metoprolol in patients undergoing thoracic surgery [24]. A meta-analysis by Chen et al. in 2019 showed that in patients undergoing cardiac surgery, being treated with RAAS inhibitors did not decrease the incidence of POAF, nor that of adverse cardiovascular events [70]. The use of RAAS inhibitors in the surgical setting is also limited by the fact that their preoperative administration for prophylactic purposes appears to be associated with postoperative acute kidney injury (AKI) [71].

### 6.5. Magnesium

Correction of plasma electrolytes is a potential preventive strategy of POAF. In particular, magnesium has been the subject of study. Bolus followed by continuous magnesium infusion during the intensive care unit stay after cardiac surgery was shown to be beneficial in a pilot study by Osawa et al. [72]. On the contrary, in a study by Solomon et al. in patients undergoing CABG, magnesium in combination with propranolol was not shown to further reduce the incidence of POAF compared to propranolol alone but significantly increased the incidence of hypotensive episodes [73]. Two meta-analyses confirmed the efficacy of magnesium in reducing POAF incidence after cardiac surgery [74,75]. Regarding the prevention of POAF with magnesium after non-cardiac thoracic surgery, as far as we know, the only available study, which showed the efficacy of this electrolyte, was carried out by Terzi et al. in 1996 [76].

### 6.6. Anti-Inflammatory Drugs

Anti-inflammatory drugs have also been investigated for POAF prophylaxis, given the unequivocal association between systemic inflammation and this arrythmia. A systematic review by Marik et al. in patients undergoing cardiac surgery demonstrated the efficacy of moderate-dose corticosteroids in reducing its incidence [77]. Imazio et al. also investigated the use of colchicine for POAF prevention, but limited to patients undergoing cardiac surgery, hence exposed to direct pericardial manipulation. Two studies by the same author in this regard had conflicting results [78,79]. Moreover, clear data on the preventive action of this drug in the setting of thoracic surgery are lacking.

### 6.7. Other Drugs

Several other pharmacological prophylactic strategies have been attempted to prevent POAF, but the results are contrasting. Dexmedetomidine was found to be effective in cardiac surgery in a meta-analysis of 13 trials by Liu et al. published in 2019 [80]. This result was not confirmed by a subsequent randomized placebo-controlled trial by Turan et al. [81]. Olprinone, a phosphodiesterase 3 inhibitor, was found to be effective in a study by Nojiri et al. in patients undergoing pulmonary resection [82]. This finding has not been confirmed by other studies to date. Perioperative statins administration in patients undergoing cardiac surgery were associated with a decrease in POAF incidence, length of in-hospital stay, and postoperative C reactive protein levels in a meta-analysis by Razaei et al. [83]. A meta-analysis by Zheng et al. confirmed these results. [84] In a study by Amar et al. in patients undergoing pulmonary resection, a trend towards POAF and other postoperative complications reduction in patients treated with atorvastatin emerged, but statistical significance was not reached [85]. Digoxin for POAF prevention was tested in many trials showing a harmful potential instead of being beneficial, and, by consequence, it is contraindicated for this purpose [20].

### 6.8. Studies Comparing the Different Preventive Strategies

Several studies comparing different prophylactic strategies have been carried out. Auer et al. demonstrated that sotatol alone or amiodarone combined with metoprolol was superior to metoprolol alone in reducing POAF incidence in patients undergoing cardiac surgery [86] In a study by Solomon et al., amiodarone was compared with propranolol in patients undergoing cardiac surgery, and the former resulted significantly more effective [87]. Another meta-analysis by Burgess et al. showed the effectiveness of both beta-blockers and amiodarone, but only the latter was able to reduce the incidence of stroke. [88] Regarding lung surgery, a meta-analysis by Riber et al. showed that amiodarone was the most effective in reducing POAF incidence [63]. In the already mentioned meta-analysis by Zhu et al., instead, no significant difference emerged between beta-blockers and amiodarone for POAF prevention in patients undergoing cardiac surgery [65].

### 6.9. What Do Guidelines Suggest

The 2014 AATS “*Guidelines for the prevention and management of peri-operative atrial fibrillation and flutter (POAF) for thoracic surgical procedures*” divide patients into three risk classes: low, medium, or high. For all patients, regardless of the class to which they belong, the guidelines recommend not to suspend the therapy with beta-blocking agents when already ongoing, and the correction of magnesemia where necessary. In intermediate-risk patients, they recommend POAF prevention with diltiazem in patients not already on beta-blockers and the postoperative administration of amiodarone. In high-risk patients, the excision of the left atrial appendage can also be considered [20] (Figure 4).

The latest 2020 ESC guidelines on atrial fibrillation distinguish patients undergoing cardiac surgery from those undergoing non-cardiac surgery. Since in a large trial, metoprolol given for prophylaxis was associated with an increased risk of death in non-cardiac surgery patients, the preventive use of beta-blockers in this specific group of patients is not recommended, while it remains indicated for patients undergoing cardiac surgery in order to reduce POAF incidence. Amiodarone is considered equally effective as a beta-blocking agent in both types of surgery [89].

## 7. Anticoagulation Management

The pharmacological treatment of POAF is a very complex and broad chapter, given the considerable variability of clinical pictures that the clinician may face, and lies outside the scope of this review. Therefore, the reader is referred to the reference guidelines.

However, the management of anticoagulation in this setting still deserves mention.

When POAF occurs, different possible scenarios may arise: rapid recovery to sinus rhythm within 48 h, recovery after 48 h, and persistence of the arrythmia. Taking decisions regarding anticoagulation, and in particular, regarding its duration, poses some difficulties. Given the very high incidence of this arrhythmia after surgery and considering its triggered nature, there is often a tendency in clinical practice not to prolong anticoagulation beyond one month from the restoration of sinus rhythm. In cancer patients, this trend is even greater due to concerns about the increased risk of bleeding. The introduction of direct oral anticoagulants, eliminating the need for recurrent International Normalized Ratio monitoring, could encourage greater recourse to long-term anticoagulant therapy in these patients as well.

It must be said that, to date, the evidence regarding the usefulness of long-term anticoagulation in patients with POAF is still conflicting. The few studies available are predominantly retrospective, while prospective ones are scarce, as well as data relating specifically to cancer patients undergoing thoracic surgery.

A study carried out in 2011 by Makhija et al. in patients developing POAF after thoracic surgery found that stroke paradoxically had a significantly higher incidence in anticoagulated patients (with warfarin or heparin) than in non-anticoagulated patients. The anticoagulated patients also suffered more bleeding episodes and other complications. A direct correlation was found between CHA_2_DS_2_ score and the incidence of stroke; however, this score did not vary significantly between the two groups, as well as overall mortality. It should be noted that the two populations were not perfectly comparable as they were different in sex percentage and in the incidence of pulmonary hypertension, HF, and peripheral vascular disease, which was higher in the anticoagulation group. The assignment of patients to the two arms was not randomized [90].

More recent evidence, instead, seems to support the use of anticoagulants [91,92]. From a systematic review by Yao et al., in fact, in 2021, it emerged that the long-term use of oral anticoagulants for the prevention of thromboembolic events was effective in patients undergoing cardiac surgery [93].

The 2014 AATS “*Guidelines for the prevention and management of peri-operative atrial fibrillation and flutter (POAF) for thoracic surgical procedures*” recommend, for POAF lasting more than 48 h, to manage anticoagulation in a similar way to that of non-surgical patients. The initiation of anticoagulation within 48 h of POAF onset should be considered based on the patient’s CHA_2_DS_2_-VASc risk score weighted against the postoperative bleeding risk. Continuation of anticoagulant therapy for at least 4 weeks after the restoration of sinus rhythm is recommended. With regard to long-term oral anticoagulation, it is indicated in the case of CHA_2_DS_2_-VASc ≥ 2, previous stroke occurrence, or in the presence of mildly or moderately impaired renal function. The recommendation to prolong anticoagulation based on CHA_2_DS_2_-VASc score also applies to patients with persistent or POAF or recurrent episodes. However, it is specified that the ideal duration of anticoagulation following POAF, at that time, was uncertain due to lack of evidence. Regarding the choice of the type of anticoagulant—oral direct or indirect or parenteral—the guidelines follow the same indications as for classical AF, although there are no specific studies focused on the use of direct oral anticoagulants for POAF [20,91].

The more recent ESC 2020 guidelines for atrial fibrillation distinguish between POAF occurring after cardiac surgery and POAF occurring after non-cardiac surgery. In fact, in the first case, patients seem to be predisposed to a lower incidence of adverse events compared to that of patients with standard AF. In the second case, on the contrary, the incidence of thromboembolic events is identical to that of classical AF. However, there is evidence that, although cardiac surgery-related POAF carries a lower risk, long-term anticoagulation may still be beneficial in this setting, as treated patients have a significantly lower risk of thromboembolism than untreated patients. Long-term oral anticoagulation is, therefore, recommended in class IIa in patients with POAF after non-cardiac surgery at risk of stroke and is still indicated, but in class IIb, if POAF occurs after cardiac surgery [89].

## 8. Conclusions

POAF is a frequent complication of surgery, including thoracic surgery for lung cancer. Since its occurrence has a close temporal and causal link with the surgical procedure, researchers have focused on understanding similarities with and differences from spontaneous AF in terms of prognostic implications and treatment modalities; likewise, studies have been focused on finding methods to predict and prevent POAF. Much of the available evidence comes from studies on cardiac surgery patients. However, there is no shortage of those carried out in the lung cancer population. Although some differences in terms of pathophysiological mechanisms and thrombo-embolic risk have emerged between AF occurring after cardiac surgery and AF occurring after thoracic surgery, the prevention strategies and the management of the anticoagulant therapy that have proved to be effective are roughly overlapping. Perioperative administration of amiodarone or beta-blockers and plasma electrolytes correction have proven to be the most effective prevention strategies; however, recent guidelines do not recommend beta-blockers initiation in naïve patients who need to undergo thoracic surgery, given the increased postoperative mortality found in this specific group of patients in clinical trials. Various risk factors, age above all, and some predictors, such as NT-proBNP, allow us to identify patients at the highest risk of developing POAF, who could benefit most of preventive therapies, avoiding their unnecessary utilization, related costs and side effects in low-risk patients. With regard to anticoagulation, the high incidence of thromboembolic events in patients with POAF suggests the correctness of following management similar to that adopted for spontaneous atrial fibrillation based on CHA₂DS₂-VASc.

## Figures and Tables

**Figure 1 cancers-13-04012-f001:**
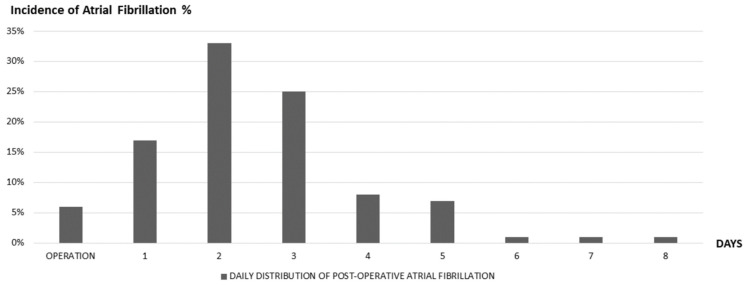
Percentage distribution (%) of AF episodes on the total of those recorded per PO day [6].

**Figure 2 cancers-13-04012-f002:**
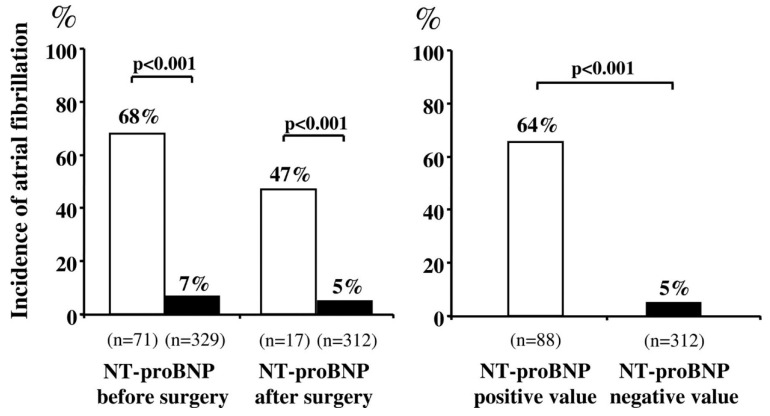
Left: POAF incidence in patients with high (open bar) versus normal (solid bar) preoperative and postoperative NT-proBNP values. Right: POAF incidence in patients with high (open bar) versus normal (solid bar) preoperative or postoperative (pooled) NT-proBNP values [23].

**Figure 3 cancers-13-04012-f003:**
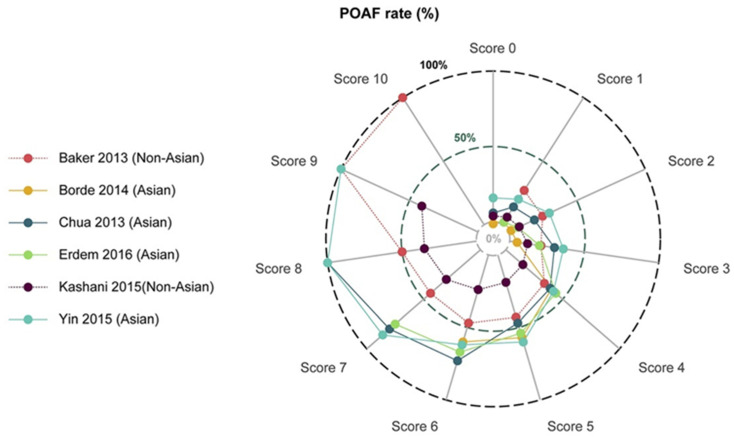
POAF incidence corresponding to each CHA_2_DS_2_-VASc score according to various trials carried out in Asian and non-Asian patients [44].

**Figure 4 cancers-13-04012-f004:**
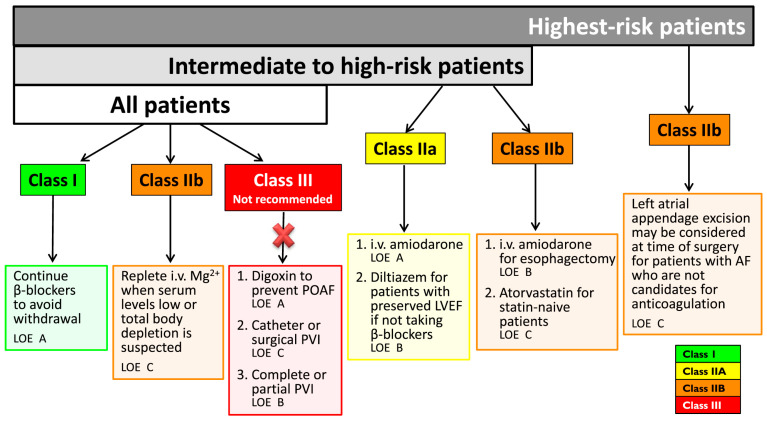
Indications regarding prophylaxis by AATS “*Guidelines for the prevention and management of peri-operative atrial fibrillation and flutter (POAF) for thoracic surgical procedures*” [20].

**Table 1 cancers-13-04012-t001:** POAF related stroke and mortality risk increase in the short and long term.

Author	Year	Number of Studies/Patients	Short Term (1 Month) Stroke Risk Increase	Short Term (1 Month) Mortality Risk Increase	Long Term Stroke Risk Increase	Long Term Mortality Risk Increase
Lin et al. [13]	2019	35/2,458,010	×1.6	×1.4	×1.4	×1.4
Koshy et al. [14]	2019	14/3,536,291	-	×3	×2.5	-
AlTurki et al. [15]	2020	28/2,612,816	×3	-	×4	-
Albini et al. [16]	2021	8/3,718,587	-	-	×4	×3.6

**Table 2 cancers-13-04012-t002:** POAF risk per type of thoracic surgery procedure. Modified from Fabiani I. et al. [6].

Title	Low Risk	Intermediate Risk	High Risk
Procedure	Flexible bronchoscopy Tracheal stenting Thoracostomy Pleurodesis Tracheostomy Rigid broncoscopy Mediastinoscopy Toracoscopic wedge resection	Simpathectomy Segmentectomy	Pleurectomy Lobectomy Lung transplantation Fistula repair Bullectomy Pneumonectomy Tracheal resection Anterior mediastinal resection
POAF incidence	<5%	5–15%	>15%

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
