# Peer review of "Atrial Fibrillation after Lung Cancer Surgery: Prediction, Prevention and Anticoagulation Management"

_cancers, 2021, doi:10.3390/cancers13164012_

Round 1

Reviewer 1 Report

This manuscript entitled Atrial fibrillation after lung cancer surgery: prediction, prevention and anticoagulation management reviews this peculiar form of arrhythmia and focuses on studies involving lung cancer patients undergoing thoracic surgery. This review paper is well organized and comprehensively described.  However, there are some suggestions for authors:

  1. Nowadays it is widely accepted that POAF is triggered by multiple factors including the pre-existing conditions or surgery-related stress. In the manuscript, as shown in Figure 1 most cases of POAF occur between the 2nd and 4th postoperative day and the incidence varies according to the extent of lung resection, the more extensive resection of lung the higher risk of POAF (Reference 6).  Regarding to this surgical characteristics, suggest the authors to discuss more detail  in the Pathophysiology paragraph about the possible primary factor contributing to the  peak AF episodes on day 2 and day 3 which might be important for the prevention of POAF in the future.

     2. Suggest to delete 6.5. Magnesium paragraph at Page 11, which is not             relevant to this report.

     3. In manuscript Page 11,  is there any difference between 6.6. Anti-                   inflammatory drugs and 6.7. Anti-inflammatory drugs?

Author Response

The authors express their gratitude for the positive comments of the Reviewer.

Our response to his/her suggestion is as follows:

  1. We added a recent study by Seitlinger et al. concerning video-assisted minimally invasive surgery which showed how the type of surgery and the surgeon's expertise are also relevant in determining the incidence of POAF.

  1. Since a study on the use of magnesium for the prevention of POAF was carried out also in patients undergone thoracic surgery in 1996, rather than eliminating the paragraph, we decided to add a few lines specifying the results of the same study.

  1. We made a mistake by re-writing the title anti-inflammatory drugs, while the correct title of paragraph 6.7 is "other drugs". Thanks for showing us the error, we have changed the title.

Reviewer 2 Report

Well done and interesting paper to read and to know

about 

Author Response

The authors express their great gratitude for the positive comments of the Reviewer.

Reviewer 3 Report

Recently I have received a paper for a review: “Atrial fibrillation after lung cancer surgery: prediction, prevention and anticoagulation management”. The paper addresses an important issue of common complications after lung cancer surgery. The review is well written and is easy to understand. I do not have significant remarks concerning the scientific style. The paper has a high didactic potential. Despite it is not a systematic review I believe it may be important to clinicians and scientists. I think it has also a citing potential.

Major remarks.

The section concerning the prophylaxis is detailed and I appreciate its didactic value. On the other hand, the vibrant issue of post-POAF anticoagulation therapy is described in a less precise way. Please consider more detailed information about the types of anticoagulative drugs recommended or state lack of these recommendations.

Minor remarks.

Line 69. Please add “non-cardiac” thoracic surgery.

Line 141. I would rather use the term sympathectomy instead “simpaticectomy”.

Figure 4. Quality is low. Please consider improving quality.

Author Response

The authors express their gratitude for the positive comments of the Reviewer.

We have re-edited the paragraph on anticoagulation making it clearer and more complete.

We have also modified what was observed in minor remarks.